# Implementation of global health competencies: A scoping review on target audiences, levels, and pedagogy and assessment strategies

**Meike Schleiff[1]** *, **Bhakti Hansoti[2], Anike Akridge[3], Caroline Dolive[3], David Hausner[3], Anna Kalbarczyk[1], George Pariyo[1], Thomas C. Quinn[2], Sharon Rudy[3], Sara Bennett[1]**

**1** Johns Hopkins School of Public Health, Baltimore, MD, United States of America, **2** Johns Hopkins School of Medicine, Baltimore, MD, United States of America, **3** Public Health Institute, Washington, DC, United States of America

* mschlei4@jhu.edu

**Data Availability Statement:** All relevant data are within the manuscript and its Supporting Information files.

## Abstract

### Background

As the field of global health expands, the recognition of structured training for field-based public health professionals has grown. Substantial effort has gone towards defining competency domains for public health professionals working globally. However, there is limited literature on how to implement competency-based training into learning curricula and evaluation strategies.

### Objectives

This scoping review seeks to collate the current status, degree of consensus, and best practices, as well as gaps and areas of divergence, related to the implementation of competencies in global health curricula. Specifically, we sought to examine (i) the target audience, (ii) the levels or milestones, and (iii) the pedagogy and assessment approaches.

### Sources of evidence

A review of the published and grey literature was completed to identify published and grey literature sources that presented information on how to implement or support global health and public health competency-based education programs. In particular, we sought to capture any attempts to assign levels or milestones, any evaluation strategies, and the different pedagogical approaches.

### Results

Out of 68 documents reviewed, 21 documents were included which contained data related to the implementation of competency-based training programs; of these, 18 were peer-reviewed and three were from the grey literature. Most of the sources focused on post-graduate public health students, professional trainees pursuing continuing education training,

**Funding:** SB and TCQ The STAR project is supported through Cooperative Agreement No. 7200AA18CA00001 by the United States Agency for International Development (USAID). The funders had no role in study design, data collection and analysis, decision to publish, or preparation of the manuscript.

**Competing interests:** The authors have declared that no competing interests exist.

and clinical and allied health professionals working in global health. Two approaches were identified to defining skill level or milestones, namely: (i) defining levels of increasing ability or (ii) changing roles across career stages. Pedagogical approaches featured field experience, direct engagement, group work, and self-reflection. Assessment approaches included self-assessment surveys, evaluations by peers and supervisors, and mixed methods assessments.

## Conclusions

The implementation of global health competencies needs to respond to the needs of specific agencies or particular groups of learners. A milestones approach may aide these efforts while also support monitoring and evaluation. Further development is needed to understand how to assess competencies in a consistent and relevant manner.

## Introduction

Global health is defined by Koplan et al. [1] as "an area for study, research, and practice that places a priority on improving health, and achieving equity in health for all people worldwide" [1]. Global health addresses the root causes of disease through an interdisciplinary and population-based effort, as well as an individual care perspective [1]. In recent years, the role of global health professionals has evolved in response to the needs of an interconnected world, from a disease-oriented and predominantly curative focus towards promoting holistic interventions which more appropriately address sociocultural influences on health, promote equity, and respond to complex societal needs [2]. Global health professionals work across many contexts and include public health workers dealing with health and its related influences and determinants in low resource settings, those supporting policy-making, medical and treatment guidelines development, budgeting and financing, service provision, data management and use, training and capacity building, and other aspects of public health programming [3,4]. We recognize that terminology of whether to call these workers "public health" or "global health" professionals likely depends on one's perspective, with many institutions in the global north defining the field as "global health" while similar roles in the global south, may be identified as "public health". In this paper, we use the term global health, recognizing the northern origins of the term [5]. Training of these diverse participants engaged in the field of global health needs to extend beyond disease-specific and other technical knowledge to include a skill base that matches the challenges of working in an interdisciplinary, cross-cultural, multi-sectoral environment to improve health outcomes worldwide [6–10].

The demand and need to define success and guide successful training in global health has led many groups, institutions, and professional societies to define sets of global health competencies. Competency-based education aims to move away from traditional learning assessment approaches—such as counting hours spent learning—to capturing the "knowledge, skills, and attitudes [or abilities] required for an acceptable level of practice" [11]. This approach opens up opportunities to focus on job performance, as well as allowing for the definition of levels of skill for assessment of progress [12,13]. Competency-based training has gained popularity in recent decades across many disciplines, including education, medicine, public health, and global health [13–15].

In global health, agencies and consortia such as the Association of Schools and Programs in Public Health (ASPPH), the Consortium of Universities for Global Health (CUGH), and

Global Health Education Consortium (GHEC), have developed tailored sets of competencies to match their specific areas of expertise and target audiences of learners [16–19]. The CUGH competency set was developed based on a review of the existing literature and thirty professional society and organization websites. On the other hand, the ASPPH utilized a multi-stage Delphi process to develop their competency set [18,20]. The United States Agency for International Development (USAID) and the Public Health Foundation (PHF) have also developed tailored sets of competencies for the different public health workforce roles in their respective organizations [7,21–24]. In an attempt to document the various approaches, a review of global health competencies published in 2017 examined 13 documents that included competency domains and proposed a set of competencies closely resembling the CUGH competency domains. This framework captures both the public health technical skills as well as "soft skills"/"leadership skills" that are applicable across the range of public health and global health roles [11].

While broad consensus is being reached at the stage of defining competency domains for the fields of public health and global health, it is significantly more difficult to decipher how training programs, institutions, and organizations are incorporating global health competencies into their learning and performance activities [25]. Many competency domains mention specific technical skills or knowledge areas, for example, within the domain of program management, competency is defined as the "ability to design, implement, and evaluate global health programs to maximize contributions to effective policy, enhanced practice, and improved and sustainable health outcomes" [18]. While such descriptions provide a broad overview of what each domain should contain, these competencies are often too general to design learning activities and track achievement of specific standards or learning objectives by a learner. Furthermore, there is now an increasing desire to codify and track levels of achievement and quantify assessment of global health practice [26].

Several groups have worked on developing structured assessment approaches across public health competencies [27]. In 2003, the Public Health Foundation's Council on Linkages Between Academia and Public Health Practice developed a three-tiered model to assess a range of public health skills [23]. Others have also developed surveys to assess communications, leadership, and analytical skills among public health professionals [28]. Another group developed a strategy to assess skills such as mobilizing partnerships and enforcing laws and regulations among public health nurses in Illinois [29]. Most recently, initiatives have begun to identify how these competencies can be applied to professionals in low and middle-income countries (LMICs) [30].

In this paper, we describe the current approaches to implementing global health competency-based education that has been developed and discuss the opportunities and needs for further development. Specifically, we seek to examine (i) the target audience, (ii) the levels or milestones, and (iii) the pedagogy and assessment approaches.

## Materials and methods

### Scope and approach

In this paper we include literature from training for professional in two fields; "Public Health" and "Global Health." While both fields have different origins and emphases, they overlap in terms of training content and competency needs, as well as a growing imperative to work closely and seamlessly together in teams and across agencies. Further, individuals with both public and global health backgrounds may fill similar jobs and roles in many agencies, particularly in international organizations [31–33]. Lastly, professionals from both fields have been working to identify approaches to implement competency domains into their respective

curricula [11,18,34]. We employed a scoping review methodology given the broad nature of our search, and a lack of a focused question but rather a need to capture a breadth of knowledge related to the implementation of competency-based curricula for these professionals [35].

## Search strategy

We searched PubMed, Embase, ERIC, and Google/Google Scholar for articles and documents published from 2003 through September 15, 2019. This timeframe was chosen because it was expected to capture the majority of the literature on competencies in public health and global health [11], and many of the key articles defining the field of global health were published in the 2003–2007 period [1,36,37]. The search encompassed four concepts in total (Table 1), which included global health and public health, education/capacity building; competency; milestone or level. We searched broadly for initiatives and studies looking at how public health and global health competency sets are being utilized and evaluated, including those from high-resource settings aimed at training professionals to work globally, including across low- and middle-income settings. Terms were identified inductively from the search results that could enable us to develop more targeted searches, and so we added terms to the search strategy (Appendix 1). We adapted the search strategy for each database to minimize the possibility of missing relevant materials. We also reviewed citations of the relevant documents that we identified.

Articles were included if they presented discrete global health competencies or the implementation of a competency-based global health training program. Conversely, articles were rejected if they did not contain data on the implementation or evaluation of competency-based curricula–i.e. they may have proposed a set of domains to cover in a curriculum, but did not provide supporting information about the learners, curriculum delivery, or assessment approaches.

## Data charting and synthesis

All included articles were reviewed in their entirety to understand in-depth the experience and status of competency-based assessment approaches in the field of global health. We captured specific data on target audiences, models for defining skill level, pedagogy, and assessment approaches and prepared a matrix, which was then refined into Tables 2 and 3.

## Results

The scoping review identified 68 documents that presented data on a competency-based training program for global health professions, which were reviewed in their entirety. Of these, 21 documents are featured in this comprehensive review (Tables 2 and 3). Of these, 18 were peer-reviewed published articles and three were other documents that included reports, policy guidelines, and electronic versions of tools. The documents had publication dates spanning

**Table 1. Concepts and specific terms utilized in the literature search.**

| Concept | Search Terms |
|---|---|
| 1. Global Health AND | global health OR international health OR one health OR public health |
| 2. Education/capacity-building AND/OR | education* OR training* OR university* OR curriculum* OR college* OR capacity OR workshop OR mentor |
| 3. Competency AND | Competency* OR skill* OR outcome* OR objective* |
| 4. Milestone AND/OR assessment | level OR layer OR matrix OR ladder OR continuum OR milestone AND/OR assessment OR evaluation |

**Table 2. Non-clinically oriented competency frameworks use and target audiences; levels and milestones; and pedagogy and assessments.**

| Article/Report | Competency Framework | Target Audiences | Levels and Milestones | Pedagogy and Assessments |
|---|---|---|---|---|
| Cole *et al.* [51] | CCPHC (Core Competencies of Public Health in Canada), CCGHR (Canadian Coalition for Global Health Research [38] (Canada) | Post-graduate students seeking careers in public health or global health | Proposes complementary milestones for global health practitioners planning to work in practice or research settings to achieve during post-graduate training. | Recommended pedagogical approaches include active reflection, direct engagement with diverse stakeholders to analyze challenges, and seeking mentorship in areas of interest. No assessment tools developed or discussed. |
| Gruppen *et al.* [59] | None cited. | Health professionals | Cites and illustrates Miller's Pyramid, which includes four levels from 1) knows, 2) knows how, 3) shows, to 4) does | Illustrates the diversity of pedagogical and assessment approaches that could be utilized depending on the competency.<br>• Pedagogy: emphasizes use of simulation, small group work, and self-directed exploration and application.<br>• Assessments include working with standardized patients, oral or written examples, and supervised practice. |
| Jogerst *et al.* (CUGH) [18] | Developed own for this paper (USA-based) | Global health trainees from a variety of disciplines | • Level I. Global Citizen Level, focused on awareness of global health among post-secondary students<br>• Level II. Exploratory Level, focused on students considering a future in global health<br>• Level III. Basic Operational level, with two sub-levels differentiating between clinicians and discipline-specific professionals working in global health and those working on managing global health programs<br>• Level IV. Advanced: plans long-term engagement in global health with leadership positions | Mentions the need for further dialogue and work in this area. |
| Sharma, *et al.* [60] | Public health Foundation (PHF)/ Council on Linkages Between Academia and Public Health Practice [23] (India) | MPH students | Relates to PHF levels in discussion; need for further research to map to these levels to specific competencies or milestones. | Mentions the need for further dialogue and work in this area. |
| Afya Bora Fellowship [39] | Developed own [39] (Sub-Saharan Africa) | Post-graduate health professionals planning to lead and manage programs in Africa | Utilizes four skill levels on a Likert scale from "weak" to "excellent." | Use of online modules (lectures, discussion) as well as in-person intensive, group-oriented sessions. Self-reported survey administered at the beginning and end of each module of the curriculum. |
| Winskell, *et al.* [40] | ASPPH [40] (USA-based) | MPH students | Specific goals to be achieved by the completion of the MPH degree program. | Use of case studies, proposal development, group discussion. |
| ASPPH [41] | ASPPH [41] (USA-based) | MPH students | Specific milestones for completion of MPH are included across all competency domains. | Not discussed beyond standard approaches to graduate education including lectures, group work, and writing papers. No assessment tool developed, nor scale discussed. |
| USAID [22] | USAID [22] (USA-based) | USAID employees | Three levels of public health competence: Basic, Intermediate, and Advanced are described and relate to kinds of roles held. | Provide online and in-person trainings as well as access to other professional development opportunities. Assessment in the form of evaluations of trainings as well as performance reviews and feedback from onsite managers. |

(*Continued*)

**Table 2.** (Continued)

| Article/Report | Competency Framework | Target Audiences | Levels and Milestones | Pedagogy and Assessments |
|---|---|---|---|---|
| Eichbaum [26] | CUGH [18] and ACGME [42] (USA-based) | Students and trainees from high resource-settings working in low-resource contexts | Not discussed. | Advocates for differentiation between acquired knowledge and skills (individual) and participatory knowledge and skills (collective) in evaluation approaches. Recommends "self-directed assessment" to evaluate these by incorporating feedback from multiple sources, including faculty, health system, and self. No tool developed. |
| Sawleshwarkar & Negin [11] | Defined set similar to CUGH [18] (USA-based) | Post-graduate public health students | Defined "key elements" in the form of knowledge/skills for each competency domain to be obtained by the end of a training program. | Not discussed explicitly; no tools developed or tested. |
| Hamer, *et al.* [43] | Developed own [43] | Mentors in LMICs who are involved in global health research | Not mentioned separately from assessment approaches. | Assessment approaches included self-reporting, monitoring mentor products from research (proposals, publications, etc.), review of mentee products, mentee satisfaction and feedback, obtaining funding, and frequency of meetings with mentees. |
| Hobson, *et al.* [44] | Council on Education for Public Health (CEPH) [44] (USA, Canada, Lebanon, Mexico, and West Indies) | MPH students | Specific qualifications for achievement of evaluation related CEPH competencies by the end of MPH degree program. | Pedagogy and assessment approaches included lectures, readings, paper-writing, group projects, field evaluations, and journaling. |

from 2011–2019; older materials that were identified in the search were either solely focused on US-based public health professionals or did not include any information on the implementation of competencies. We cited a number of these older documents in the introduction to this paper in order to provide some context and acknowledge prior work undertaken in this field.

Three themes were identified based on the evidence identified in the review to organize our findings and recommendations: (i) target audiences; (ii) milestones or levels; and (iii) pedagogical and assessment strategies and are presented sequentially in the remainder of the results section. These topics serve as the organizing framework for this section. Tables 2 and 3 provide an overview of the findings across each topic, organized chronologically by year of publication. Table 2 focuses on findings that are non-clinically oriented or multi-disciplinary programs, whereas Table 3 focuses on clinically oriented programs. We have separated these two categories in order to be able to compare across programs focusing on public health professionals versus those providing global health content to other health professionals, and the inherent nuances of training these two groups of professionals.

## Target audiences for global health competencies

This scoping review identified three major groups of target audiences for public health and global health competencies: post-graduate public health students, professional development for global health workers, and global health training for clinical and allied health professionals. The competencies for the first audience included U.S.-based training programs preparing individuals for public health careers internationally while the second category focuses more on training programs for LMIC-based public health professionals. The last category includes individuals who may be from the US but ultimately who plan to work or already work in a low-resource health setting as a clinical or allied health professional.

**Table 3. Clinically oriented competency framework use and target audiences, levels and milestones, and pedagogy and assessments.**

| Article/Report | Competency Framework | Target Audiences | Levels and Milestones | Pedagogy and Assessments |
|---|---|---|---|---|
| Redwood-Campbell, et at. [54] | CanMEDS [45] (Canada) | Medical students and residents planning careers in public health or global health | Not discussed beyond targets or skills to achieve by the end of the medical education program. | Advocates for evaluation of service learning, field placements utilizing self-reflection, group learning, simulation, and apprenticeship. No assessment tools developed or proposed. |
| Gladding, et al. [62] | ACGME [42] (USA-based) | Pediatric residents participating in an international elective. | Not discussed beyond targets or skills to achieve by the end of the medical education program. | Reflective essays were utilized to qualitatively evaluate progress towards ACGME domains as well as clarify personal goals and values. |
| Veras, et al. [46] | CanMEDS [46] (Canada) | Occupational therapy and physiotherapy students studying global health | None specifically mentioned; the study explored existing knowledge and skills as well as learning needs across competencies. | Online assessment survey included 3-point scale of global health knowledge, a 5-point scale of global health skills, and a 5-point scale of global health learning needs. |
| Munyewende, et al. [64] | WHO [47] and ICN (International Council of Nurses) managerial competencies [48] (South Africa) | Clinic nursing managers working in public health programs in South Africa. | Not discussed beyond general reference to expected skill levels for nurse managers. | Manager self-assessment and assessment by subordinates. 360-degree competency evaluation tool developed with questions on a 10-point scale of increasing skill level. |
| Wroe, et al. [58] | ASPPH [41] (USA-based) | Internal medicine residents participating in global health training | Not discussed. | Emphasis on opportunities to engage in real-life practice or simulations. Assessment instrument is a student interview-based tool (scenarios). Designed for use in evaluating portions of residency programs, job candidate evaluation, and ongoing practical trainings. |
| Knight, et al. [61] | HPCSA (Health Professions Council of South Africa) [49] (South Africa) | Clinicians receiving public health training | Measured whether specific outcomes or skills were acquired or achieved by the end of the program. | Pedagogy includes field placements working on community diagnosis and planning and evaluating a program. Assessment was a student online survey covering specific skills within each competency domain. The survey used a Likert scale with agree/neither agree nor disagree/disagree options for whether learners has acquired that skill. It also allowed students to provide qualitative feedback on their experience. |
| Kim, et al. [52] | None mentioned. (South Korea) | Graduate nursing students studying global health | Not discussed. | Advocates for going beyond didactic lessons and incorporating simulations and field-based scenarios. Assessment approach was the application of Veras, et al. [57] tool. |
| Douglass, et al. [34] | CUGH Expert Working Group (EWG) [18] (USA-based) | Emergency medicine residents studying global health | • Level 1: Focus on awareness and knowledge<br>• Level 2: Focus on understanding and describing<br>• Level 3: Focus on participation, observation, and application<br>• Level 4: Focus on collaboration, management, and evaluation<br>• Level 5: Focus on creating, advocacy, and leadership | Suggests appropriate pedagogy and assessments for each level, with overlap between levels:<br>• Level 1: Group discussions, course assessments, simulations, essays<br>• Level 2: Group discussions, simulations, observation<br>• Level 3: Self-assessments, assessments from field experiences (360 evaluations)<br>• Level 4: Mentor evaluations, presentations<br>• Level 5: Colleague or partner evaluations, academic productivity, or curriculum development |
| Kelly & Lazenby [50] | Developed own (USA-based but reflecting global faculty and institutional perspectives) [50] | Graduate global health nursing students | Discusses expectations for nurses who have completed graduate-level global health training | Pedagogy and assessment included use of case vignettes, essays and other types of critical analysis and reflection, development of plans and proposals, supervised clinical activities in host countries, discussion and reflection with host team members. |

### Post-graduate public health students

Two articles written by a team from the CUGH sub-committee on competencies broadly address global health professionals, recognizing the diversity of backgrounds and levels of expertise of public health and global health trainees [17,18]. In Canada, multiple universities under the GHEC have engaged in individual as well as collective review and debate about global health competencies [51]. The ASPPH also reflects a university-based consensus-building process to determine competencies for post-graduate degree-seeking students across the Council on Education for Public Health (CEPH)-accredited U.S. universities [41,44].

### Global health professionals

USAID and the Afya Bora Fellowship focus on professional skill-building targeted towards specific career trajectories, preparing leaders to manage global health programs in Africa [39]. The USAID competencies are geared towards employees within the USAID system and target political and managerial competencies as well as content areas aligned with global health-related USAID strategy [22].

### Clinical and allied health professional training component

Nine articles focused on medical, nursing, pharmacy, dental, and rehabilitation students receiving targeted training in global health [34,52–56]. For example, competencies based on the GHEC domains were adapted for family medicine trainees in Canada who plan to engage in global health; these include values and "soft skills" that will enable a physician to operate effectively and appropriately in diverse contexts [54]. A specialized matrix for global oral health was also developed building on the CUGH competency domains and resulted in a list focused on dental disease-specific knowledge, including disease risk factors and a set of more general interpersonal and professional skills [55].

### Defining skill levels and "milestones" for learners

As numerous groups continue to work on defining and refining competency domains, many have evolved to recommend specific levels or milestones for trainees, which can be tailored depending on career trajectory or scope of work [51]. Tools to organize and measure learner progress and growth within competency domains have begun to emerge, particularly in the last few years (2015–2019) [11,27,29,57,58]. We identified two approaches to tracking the achievement of levels of competency across domains for professionals in public health and global health. The first approach focuses on sequential or tiered levels of ability and is more hierarchical in nature. This approach focuses more on the achievement of advancing or specialized skills. The second approach has a more longitudinal view and shows differing skills as the roles evolve (Fig 1). While the first approach is more likely to be tiered and perceived and implemented as a linear progression, the second can feasibly include parallel routes to different milestones and is more apt at placing individuals within a functional category.

   The USAID competency framework utilized a three-level approach with basic, intermediate, and advanced categories [21,22]. Similarly, Gruppen, *et al.* [59] employed a four-level model with beginner levels focusing on knowledge, and more advanced levels focus on skills. The Afya Bora fellowship measures skills on a scale from weak to excellent [39].

   Building on CUGH's set of core competencies, a four-level approach was developed by CUGH beginning with a "Global Citizen" basic awareness level for a trainee pursuing a field with bearing on global health but not necessarily with sustained or direct engagement. It ends at an "Advanced" level of student who plans to have long-term engagement in global health

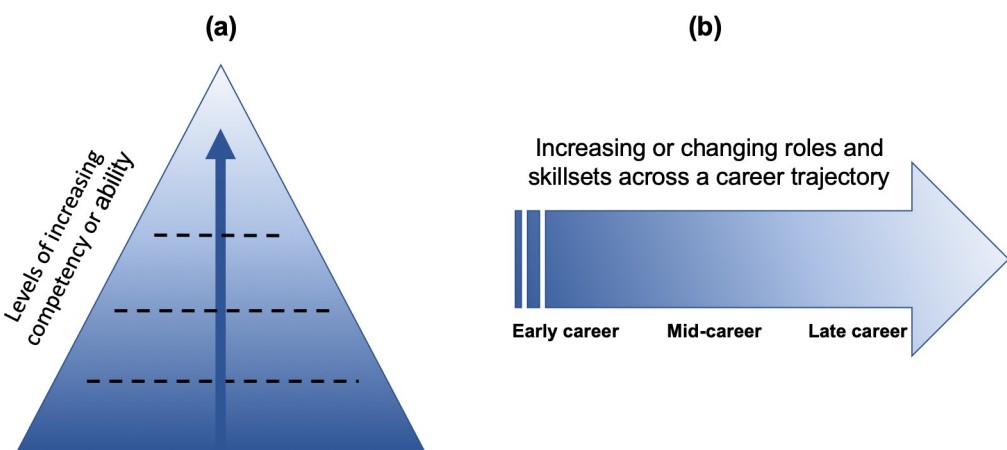

**Fig 1. Two approaches identified for conceptualizing levels of competency.**

and work towards holding substantive leadership positions [18]. The Global Emergency Medicine Think Tank Education Working Group (EWG) also used a structured process to determine milestones across five levels for emergency medicine trainees undertaking global health experiences. The levels ranged from Level 1 characterized by very basic knowledge of major concepts across competency domains to a Level 5, indicating a leader and teacher who is able to contribute to building the field of global health across one or more competency domains [34].

Most of the clinically-oriented articles (except Douglass, et al.) [34] described specific milestones or expectations to be achieved by the end of the training programs, and several of the non-clinical examples (Sharma, et al., Winskell, et al., Hobson, et al., and the ASPPH) [40,44,60] also emphasized specific levels and milestones to be achieved by the completion of an MPH program.

## Pedagogy and assessment strategies and tools

**Pedagogy.** Pedagogical approaches revealed several consistent themes across the clinical and non-clinical articles. The first key theme was direct engagement and application of learning in clinical or field settings. For clinical settings, this could include practice with standardized patients, supervised clinical activities, other engagement with host team members, or similar apprenticeship models [34,50,52,54]. For non-clinical field activities, these field placements included participation in activities from community diagnosis and program planning through to project evaluation [44,51,61]. Simulations for use in classroom settings were also a related theme found across both target audiences [34,54,58,59].

The second cross-cutting theme relates to group work. This theme included small group discussions in classroom settings, group projects, collaborative activities to develop proposals or other products, and group problem-solving in clinical contexts [34,39,40,44,50,54,59]. Eichbaum [26], in particular, noted that the role of group work may be seen differently across cultures that are more individualistic versus more collectivist, and also that learners may have different levels of experience, comfort, or expectations for how these group activities are managed. What is apparent is that while group work is identified as a core educational strategy, it may be more difficult to implement in certain cultural settings, and thus group work approaches that are facilitated or more structured may be important to adopt.

The final cross-cutting theme related to self-reflection, which was present for both clinical and non-clinical audiences as well as across levels of graduate and post-graduate training. Different articles mentioned different approaches, including reflective essays and journaling, as

well as reflections with peers as a strategy to strengthen the absorption of curricula [34,44,50,51,54,62].

**Assessment strategies and tools.**   We identified three kinds of evaluation tools that have been developed and used to assess individual competencies: i) self-assessment surveys, ii) assessment from multiple stakeholders or perspectives, and iii) mixed methods assessments using surveys, scenarios or interviews. The findings for each kind will be discussed in sequence.

Several of the articles included self-reflection approaches as a means to facilitate assessment. The Afya Bora fellowship was the only group that utilized a self-reflection tool both before and after each module of the curriculum to demonstrate a change in knowledge/skills/comfort [39]. Veras, *et al.* [63] published the results of a validated self-assessment survey developed to assess global health competencies [63] and also utilized the tool with rehabilitation students to self-assess knowledge, skills, as well as learning needs [46]. The tool begins to address the need for validated structured assessment that is able to assess gaps knowledge across different participant groups uniformly and has since been applied to global health training in South Korea with positive results as well as with rehabilitation students in Canada [46,52,63]. Gladding, *et al.* [62] and Redwood-Campbell, *et al.* [54] also emphasized the need to better evaluate student self-reflection and essays to assess progress towards milestones or other training-specific goals.

Two documents included in the scoping review utilized input from multiple perspectives, including self-evaluation [29,64]. Both used a variation of a 360-degree evaluation, or multi-rater feedback, approach [65] that included feedback from the public health professional's subordinates, colleagues, and supervisors, as well as a self-evaluation by the individual themselves. Both approaches utilized Likert scales, though one scale focused on skill development, and the other focused on the development of competency in a particular area from the role of learner to that of teacher/instructor. USAID also emphasized feedback from supervisors and other performance reviews [22], and Eichbaum [26] advocated for evaluation from multiple sources, including faculty, representatives from the health system, and self-evaluation.

Wroe, *et al.* [58] developed a series of scenarios common to global health practice in order to capture more nuanced feedback related to the "soft skills" in global health. Interviewers assessed whether respondents had received adequate global health training in order to be prepared for continued practice in areas such as professionalism, self-care, and interpersonal and cross-cultural communication [58]. Another approach by Knight, *et al.* [61] utilized a self-assessment survey that included quantitative measures as well as open-ended qualitative questions.

Douglass and colleagues [34] hypothesized that the evaluation strategy might need to be responsive to the level of the learner, and thus, the skill being assessed [34]. While earlier levels focus more on self-reflection and formal assessment processes, the more advanced levels require more objective measures of achievement, such as peer assessment or scientific publications [34].

## Discussion

The implementation of competency-based educational models is both nuanced and challenging. A variety of approaches have been adopted for different learner groups, and more may become priorities as global health training takes on greater prominence at the undergraduate level as well. However, most focus on developing specific measures for achievement, such as requirements for completing a degree program, and use a variety of evaluation strategies. The challenge of a meaningful and appropriate assessment of competence [66] has led to efforts to offer more concrete approaches through the development of levels, milestones, and evaluation tools. Based on our synthesis of the evidence in this review, we developed a box of recommendations for program implementers and priority areas for further research (Box 1) in order to provide actionable steps for others who want to continue to develop this field.

> ## Box 1. Recommendations for program implementation and further research
>
> Program Implementation Recommendations:
>
> - Supporting the global health workforce across all levels of experiences must focus not only on technical skills, but also on leadership, communication, cultural competency, etc. that support the development of impactful global health professionals.
>
> - Define levels and milestones for programs or organizations that align with desired workforce advancement and supports consistent approaches to performance review
>
> - Build opportunities for application of learning into training programs, including structured and context-appropriate experience working with peers and mentors to enable shared learning and practice in working in diverse teams
>
> - Develop appropriate assessment approaches based on level of learner, and use approaches including case examples, simulations, and 360-degree reviews to support more advanced learners
>
> Further Research Recommendations:
>
> - Develop frameworks and theories of change to support curriculum customization and measure impact, which are based on existing global health competency literature.
>
> - Build on the important initial work to define priorities for competencies that is taking place among LMIC-based global health professionals and seek to better understand learning needs of field based participants
>
> - Identify strategies to develop innovative pedagogical interventions to support the development of core competencies (aka "soft skills") and move beyond training centered around graduate public health degree programs
>
> - Develop robust measurement of long term impact of innovative pedagogical approaches, including what works best across contexts and for different kinds of learners
>
> - Study the use of assessment approaches that have been advocated for in global health training, particularly for leadership and related skills, including self-reflection and 360-degree reviews

Some recent approaches for teaching global health have been built on the foundation of competency-based medical education (CBME) from clinical settings [67]. Many of the best practices for CBME have developed specific anchors or skills that can be objectively observed or otherwise identified to define the achievement of a competency level. It is however, acknowledged that challenges remain on how to meaningfully assess "soft skills", such as leadership, communication, and cross-cultural practice, which are central to global health work [68]. Assessment of "soft-skills" may require a process-oriented approach to understand learner experiences and establish feedback mechanisms. These competencies ultimately also need to be developed at an individual, program, and broader agency or societal levels and involve a variety of stakeholders (educators, peers, supervisors) within training programs as well as post-graduation [69].

Another promising direction is the emerging focus on adapting competency frameworks and approaches to utilizing them for use in LMIC [30,52,64]. This has included assessing competence in LMICs while accounting for local learning styles, culture, and other contextual factors relevant to global health work. However challenges do exist, for example, assessment of competencies within cultures where teamwork and direct engagement are commonplace and necessary but traditional assessment approaches do not capture these skills fully [26,56] or how to account for cultural and practice setting differences when aiming to develop a globally-applicable set of competencies [56]. In the field of global health, increased emphasis is needed on the competencies related to participatory approaches, learning across disciplines and in resourceful ways, and maintaining a social justice and health equity lens [26].

Long-term capacity strengthening in LMICs to achieve the Sustainable Development Goals (SDGs) must consider the context in which global health practice occurs [70,71]. Further evidence on approaches used in LMIC settings and their effectiveness is needed. Future efforts might include systematically documenting consensus on competencies based on empirical studies that include input from a wide range of global health stakeholders, including LMIC national-level policymakers and leaders, managers, academics and researchers, and civil society. Competency-based education is held up as an approach that can make training as applied and impactful as possible. However, further rigorous evaluation of the impact—both immediate and longer-term—of global health training programs is needed. Furthermore, competency-based education for students and early-career professionals undergoing more knowledge-focused training is inherently different from the skills desired among senior health professionals. Colloquially these skills are often termed "leadership skills." Understanding the nuances of how leadership skills can strengthen public health practice and how these skills and be codified to provide focused monitoring, feedback, and training is the key to supporting the global health workforce.

Lastly there is an unspoken dynamic between the focus on "structural competency" as s as a strategy to develop learning/teaching objectives focused on delivering knowledge around determinants of health versus the use of competencies to identify the key areas of emphasis required to develop impactful global health citizens. Much of the work featured in this review focuses on the needs of the former i.e., identifying domains for teaching medical and public health students so that they understand the social determinants of health from a theoretical and empirical perspective, but not necessarily providing them with the skills to address such inequities. As the field of global health and competency-based education develop there is an opportunity for educators to use the competencies to deliver a farsighted approach to global health educational content.

Our review had several limitations, which included reliance on published literature and ongoing ambiguity around the most appropriate search terms to utilize. To the first point, although we have included published and grey literature in this review, we believe that there are more examples of implementation of competency-based curricula in LMIC contexts that what we were able to find. This links with the second point, which is that different programs, educational systems, and health professions use a range of different terms to describe both global health as well as competency-based curricula. Therefore, conducting a systematic and exhaustive search was a challenge.

## Conclusions

Global health is already a very dynamic field and is sure to change even more in the future. While many different voices are joining the debate about how the field will evolve by providing perspectives, tools, and learning activities, a great deal of work must be done to align, validate

and evolve these contributions towards translatable, actionable, and trustworthy instruments, resources, and opportunities. The capacity development needs of professionals in government versus non-governmental organizations, academic or research- versus program implementation-focused institutions, and the public or non-profit versus private sectors can vary greatly, as can the individual learning styles of professionals in those settings. Competencies and their assessment may also need to vary accordingly to respond to the needs of specific agencies or particular groups of learners. Further discussion and action on the role and implementation of competency-based education better equip the global health workforce as they address current and emerging global health challenges is needed.

## Supporting information

**S1 Checklist. Preferred reporting items for systematic reviews and meta-analyses extension for scoping reviews (PRISMA-ScR) checklist.**
(DOCX)

**S1 Appendix. Appendix I: Search terms by database.**
(DOCX)

## Author Contributions

**Conceptualization:** Meike Schleiff, Bhakti Hansoti, Sara Bennett.

**Data curation:** Meike Schleiff.

**Formal analysis:** Meike Schleiff.

**Funding acquisition:** David Hausner, Sara Bennett.

**Investigation:** Meike Schleiff, Bhakti Hansoti.

**Methodology:** Meike Schleiff, Bhakti Hansoti, Anike Akridge, Caroline Dolive, David Hausner, Anna Kalbarczyk, George Pariyo, Thomas C. Quinn, Sharon Rudy, Sara Bennett.

**Project administration:** Meike Schleiff.

**Supervision:** Bhakti Hansoti, Sara Bennett.

**Validation:** Meike Schleiff, Bhakti Hansoti, Anike Akridge, Caroline Dolive, Anna Kalbarczyk, George Pariyo, Thomas C. Quinn, Sharon Rudy, Sara Bennett.

**Visualization:** Meike Schleiff.

**Writing – original draft:** Meike Schleiff.

**Writing – review & editing:** Meike Schleiff, Bhakti Hansoti, Anike Akridge, Caroline Dolive, David Hausner, Anna Kalbarczyk, George Pariyo, Thomas C. Quinn, Sharon Rudy, Sara Bennett.

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
