## [Decision Letter · Decision Letter 0]

19 May 2020

PONE-D-19-35888

Implementation of Global Health Competencies:

A Scoping Review on Target Audiences, Levels, and Pedagogy and Assessment Strategies

PLOS ONE

Dear Dr. Schleiff,

Thank you for submitting your manuscript to PLOS ONE. After careful consideration, we feel that it has merit but does not fully meet PLOS ONE’s publication criteria as it currently stands. Therefore, we invite you to submit a revised version of the manuscript that addresses the points raised during the review process.

We would appreciate receiving your revised manuscript by Jul 03 2020 11:59PM. To enhance the reproducibility of your results, we recommend that if applicable you deposit your laboratory protocols in protocols.io, where a protocol can be assigned its own identifier (DOI) such that it can be cited independently in the future. For instructions see: http://journals.plos.org/plosone/s/submission-guidelines#loc-laboratory-protocols

We look forward to receiving your revised manuscript.

Kind regards,

Elisa J. F. Houwink, MD, PhD

Academic Editor

PLOS ONE

Journal Requirements:

Additional Editor Comments (if provided):

Reviewers' comments:

Reviewer's Responses to Questions

**Comments to the Author**

1. Is the manuscript technically sound, and do the data support the conclusions?

Reviewer #1: Yes

Reviewer #2: Yes

Reviewer #3: Partly

2. Has the statistical analysis been performed appropriately and rigorously? 

Reviewer #1: N/A

Reviewer #2: Yes

Reviewer #3: No

3. Have the authors made all data underlying the findings in their manuscript fully available?

Reviewer #1: Yes

Reviewer #2: Yes

Reviewer #3: No

4. Is the manuscript presented in an intelligible fashion and written in standard English?

Reviewer #1: Yes

Reviewer #2: Yes

Reviewer #3: Yes

5. Review Comments to the Author

Reviewer #1: This is a timely piece of work and will make a strong contribution to the literature. The Discussion and Conclusion sections are well written and informative.

The manuscript contains grammatical errors that must be corrected.

Reviewer #2: I was excited to read this article and to learn from its review of the literatures on global health education competencies. The work the authors do in surveying this literature is undoubtedly useful, and, inter alia, it also helps further consolidate claims that a global health education field exists in the first place. I also found the paper to be clearly written and accessible.

These qualities noted, I came away quite disappointed by the lack of actionable content in the article, as well as a sense of frustration with many areas of global health education (and competency) left unaddressed. In particular, it seems to me that the article fails to address the growing area of undergraduate global health education, including the many books and articles designed to teach college students about priorities and challenges in global health delivery. For example, all the work of the Harvard scholars involved in creating the influential textbook Reimagining Global Health, including their salutary emphasis on a ‘biosocial’ approach, was ignored. I also felt that wider work on cultural competency was sidelined, along with the sorts of work that is widely cited in global health on the challenges of global governance and humanitarianism amidst the inequalities of global development. At the same time, the sort of biomedical leadership taken by The Lancet’s Richard Horton in advancing new medical voices in global health was also ignored.

Put together these sorts suggest a bigger weakness in the paper: namely, its failure to articulate and discuss clearly a widespread concern in global health education with ‘structural competency’ – a term often used to describe the learning objectives involved in being able to describe the social and economic determinants of unequal or bad health outcomes globally, as well as with the social forces that delimit how global health policy is developed in response. Relatedly, a whole vocabulary of terms developed by social scientists - medical anthropologists, sociologists and geographers all included – seemed to be missing from this article’s study and discussion. Such terms include those useful for describing global and structural forms of pathogenesis associated with globalization including ‘syndemics’, ‘biological citizenship’, and ‘biological sub-citizenship’, but they also extend to debates over global health delivery in the context of structural violence and neoliberal austerity, including debates over health systems strengthening, diagonalization and the problems of exclusion associated with vertical biomedical interventions in global health. Any competent account of global health competencies ought to be engaging with the literatures that future medical students and MPH students are reading about these terms. It should in turn discuss what sorts of milestones and metrics might be developed to measure how well students develop literacy in these critical global health areas. I hope that some major revisions to this paper can address some of these missed opportunities, and, for the same reason, I hope these comments are useful.

Reviewer #3: GENERAL COMMENTS

The manuscript by Schleif and colleagues attempted a comprehensive review of expectations for competency in global health, and how the competencies have been implemented in various context. There have been many developments in global health education over that past 3 decades, and such a review would be helpful, provided that it does not contribute to the growing confusion about terminologies and expectations for faculty and trainees, and those who practice what we are preaching. For example, there is deep confusion in this article about what global health is, and what it is not, with an attempt to split the difference by advocating the conflated phrase "global public health". The differences between and among public health, international health, global health, population health, one health, etc. have been debated extensively, from the perspective of scope and competencies overlap and distinctiveness. The review of 21 articles published over the past 18 years does not capture the nuances of the debates and in its present form, falls short of revealing themes that might advance the discussion.

SPECIFIC COMMENTS

Line 19: The abstract should include information about the pool of articles from which less than a third were selected, and the selection criteria, which was not described convincingly anywhere in the manuscript.

Line 25: The abstract notes "360 evaluations" which is a confusing term, and should be explained to readers. The abstract also notes only two assessment methods, whereas three were described in the main text.

Line 37: This is an extremely broad scope that conflates core public health and medicine. Describing global health with such broad scope renders it without distinction. The debate dates back to the differentiation of international health and global health, and now population health. For this exercise to be useful, the narrative must be more precise than presented here.

Line 93: This is a very confusing terminology that conflates global health and public health. The first statement in the materials and methods section adds to the confusion.

Line 101: The introduction section should have alerted readers that the paper has two foci. Until this point, the assumption is that this is all about global health, which is differentiated from public health.

Line 104: This is not an ideal situation. The core foundational degree is typically public health, and additional training is essential to transfer skills to global health context, otherwise there is incompetence, and potential to make situations worse.

Line 108: Extremely confusing logic (Global Public Health).

Line 113: PUBLISHING LIMITED-, 14(3), pp.427-432.

Haq, C., Rothenberg, D., Gjerde, C., Bobula, J., Wilson, C., Bickley, L., Cardelle, A. and Joseph, A., 2000. New world views: preparing physicians in training for global health work. FAMILY MEDICINE-KANSAS CITY-, 32(8), pp.566-572.

Bunyavanich, Supinda, and Ruth B. Walkup. "US public health leaders shift toward a new paradigm of global health." American Journal of Public Health (2001): 1556-1558.

Fineberg, H.V., Green, G.M., Ware, J.H. and Anderson, B.L., 1994. Changing public health training needs: professional education and the paradigm of public health. Annual review of public health, 15(1), pp.237-257.

Schultz, S.H. and Rousseau, S., 1998. International health training in family practice residency programs. Family medicine, 30(1), pp.29-33.

Neufeld, V., 1992. Training in international health: a canadian perspective. A North-South Debate, p.193.

Davis, D., 1998. Global health, global learning. British Medical Journal, 316(7128), pp.385-390.

Gebbie, K., Merrill, J. and Tilson, H.H., 2002. The public health workforce. Health

Line 140: Why were the other articles rejected? Table 2 rightly includes some non-U.S. authors and context as global health is a global practice. What were the criteria for including these and perhaps rejecting others?

Line 147: How were these identified? is this a systematic methodology or eyeballing?

Page 17 (no line numbers): The abstracted noted only 2 types of assessments.

Page 17: Define "360 evaluation" for readers or provide reference.

Page 19, second paragraph: This statement about public health is likely to be met with deep resistance. I do not think that most public health professionals will accept the view that public health thinking was build on a foundation of clinical settings.

6. PLOS authors have the option to publish the peer review history of their article (what does this mean?). If published, this will include your full peer review and any attached files.

Reviewer #1: No

Reviewer #2: Yes: MATTHEW SPARKE

Reviewer #3: Yes: Oladele A. Ogunseitan

---

## [Author Response · Author response to Decision Letter 0]

28 Jun 2020

Response to Reviewers

Reviewer #1: This is a timely piece of work and will make a strong contribution to the literature. The Discussion and Conclusion sections are well written and informative.

The manuscript contains grammatical errors that must be corrected.

Response: Thank you. We have conducted another thorough copy edit. 

Reviewer #2: I was excited to read this article and to learn from its review of the literatures on global health education competencies. The work the authors do in surveying this literature is undoubtedly useful, and, inter alia, it also helps further consolidate claims that a global health education field exists in the first place. I also found the paper to be clearly written and accessible.

These qualities noted, I came away quite disappointed by the lack of actionable content in the article, as well as a sense of frustration with many areas of global health education (and competency) left unaddressed. In particular, it seems to me that the article fails to address the growing area of undergraduate global health education, including the many books and articles designed to teach college students about priorities and challenges in global health delivery. For example, all the work of the Harvard scholars involved in creating the influential textbook Reimagining Global Health, including their salutary emphasis on a ‘biosocial’ approach, was ignored. I also felt that wider work on cultural competency was sidelined, along with the sorts of work that is widely cited in global health on the challenges of global governance and humanitarianism amidst the inequalities of global development. At the same time, the sort of biomedical leadership taken by The Lancet’s Richard Horton in advancing new medical voices in global health was also ignored.

Response: Thank you; this is well noted. We acknowledge the lack of addressing undergraduate global health specifically and agree that we did not specifically delve into the challenges of advancing new medical voices. The purpose of this review was narrow, and we choose to focus on the implementation of, “global health competency-based education that has been developed and discuss the opportunities and needs for further development. Specifically, we seek to examine (i) the target audience, (ii) the levels or milestones, and (iii) the pedagogy and assessment approaches.” We have created a box of recommendations for program implementors and areas for further research based on the synthesis from this review on p. 18. 

Put together these sorts suggest a bigger weakness in the paper: namely, its failure to articulate and discuss clearly a widespread concern in global health education with ‘structural competency’ – a term often used to describe the learning objectives involved in being able to describe the social and economic determinants of unequal or bad health outcomes globally, as well as with the social forces that delimit how global health policy is developed in response. Relatedly, a whole vocabulary of terms developed by social scientists - medical anthropologists, sociologists and geographers all included – seemed to be missing from this article’s study and discussion. Such terms include those useful for describing global and structural forms of pathogenesis associated with globalization including ‘syndemics’, ‘biological citizenship’, and ‘biological sub-citizenship’, but they also extend to debates over global health delivery in the context of structural violence and neoliberal austerity, including debates over health systems strengthening, diagonalization and the problems of exclusion associated with vertical biomedical interventions in global health. Any competent account of global health competencies ought to be engaging with the literatures that future medical students and MPH students are reading about these terms. It should in turn discuss what sorts of milestones and metrics might be developed to measure how well students develop literacy in these critical global health areas. I hope that some major revisions to this paper can address some of these missed opportunities, and, for the same reason, I hope these comments are useful.

Response: Thank you. We have added this important consideration of structural competency to our discussion on p.20. See the second to last paragraph in the discussion. 

Reviewer #3: GENERAL COMMENTS

The manuscript by Schleif and colleagues attempted a comprehensive review of expectations for competency in global health, and how the competencies have been implemented in various context. There have been many developments in global health education over that past 3 decades, and such a review would be helpful, provided that it does not contribute to the growing confusion about terminologies and expectations for faculty and trainees, and those who practice what we are preaching. For example, there is deep confusion in this article about what global health is, and what it is not, with an attempt to split the difference by advocating the conflated phrase "global public health". The differences between and among public health, international health, global health, population health, one health, etc. have been debated extensively, from the perspective of scope and competencies overlap and distinctiveness. The review of 21 articles published over the past 18 years does not capture the nuances of the debates and in its present form, falls short of revealing themes that might advance the discussion.

Response: Thank you for raising this concern. We have addressed this point at the end of the paper, and we have revised our terminology to “global health” with the emphasis that there are differences in understanding this term depending on the reader’s perspective. 

SPECIFIC COMMENTS

Line 19: The abstract should include information about the pool of articles from which less than a third were selected, and the selection criteria, which was not described convincingly anywhere in the manuscript.

Response: Thank you. We have added the requested details in the abstract. 

Line 25: The abstract notes "360 evaluations" which is a confusing term, and should be explained to readers. The abstract also notes only two assessment methods, whereas three were described in the main text.

Response: Thank you. We have taken this terminology out of the abstract and defined it further in the main text. 

Line 37: This is an extremely broad scope that conflates core public health and medicine. Describing global health with such broad scope renders it without distinction. The debate dates back to the differentiation of international health and global health, and now population health. For this exercise to be useful, the narrative must be more precise than presented here.

Response: Thank you for this important perspective, and we certainly do not wish to add to confusion. We have added a sentence in the introduction (lines 49-51) to frame our use of these terms. 

Line 93: This is a very confusing terminology that conflates global health and public health. The first statement in the materials and methods section adds to the confusion.

Response: We appreciate the feedback, and we have revised our terminology to “global health” and have further defined our perspective in the introduction as well. 

Line 101: The introduction section should have alerted readers that the paper has two foci. Until this point, the assumption is that this is all about global health, which is differentiated from public health.

Response: Thank you. We have clarified our opening to the Materials and Methods section to explain that our search aimed to capture experience from both of these fields in order to inform global health training curricula going forward. 

Line 104: This is not an ideal situation. The core foundational degree is typically public health, and additional training is essential to transfer skills to global health context, otherwise there is incompetence, and potential to make situations worse.

Response: Thank you, and we appreciate your concern. We aimed to reflect the reality that professionals with clinical and public health training as well as with other backgrounds altogether may be holding positions in global health work and therefor may require training in order to ensure that they are all prepared with core skills needed for effective global health practice. 

Line 108: Extremely confusing logic (Global Public Health).

Response: Thank you. We have omitted this sentence. 

Line 113: This start date (2003) for the review is inadequate because the review missed critical milestones in the definition of global health and its distinction from public health and international health. Particularly regarding the re-training of health professionals. For example, see:

Urkin, J., Alkan, M., Henkin, Y., Baram, S., Deckelbaum, R., Cooper, P. and Margolis, C.Z., 2001. Integrating global health and medicine into the medical curriculum. EDUCATION FOR HEALTH-ABINGDON-CARFAX PUBLISHING LIMITED-, 14(3), pp.427-432.

Haq, C., Rothenberg, D., Gjerde, C., Bobula, J., Wilson, C., Bickley, L., Cardelle, A. and Joseph, A., 2000. New world views: preparing physicians in training for global health work. FAMILY MEDICINE-KANSAS CITY-, 32(8), pp.566-572.

Bunyavanich, Supinda, and Ruth B. Walkup. "US public health leaders shift toward a new paradigm of global health." American Journal of Public Health (2001): 1556-1558.

Fineberg, H.V., Green, G.M., Ware, J.H. and Anderson, B.L., 1994. Changing public health training needs: professional education and the paradigm of public health. Annual review of public health, 15(1), pp.237-257.

Schultz, S.H. and Rousseau, S., 1998. International health training in family practice residency programs. Family medicine, 30(1), pp.29-33.

Neufeld, V., 1992. Training in international health: a canadian perspective. A North-South Debate, p.193.

Davis, D., 1998. Global health, global learning. British Medical Journal, 316(7128), pp.385-390.

Gebbie, K., Merrill, J. and Tilson, H.H., 2002. The public health workforce. Health

Response: Thank you. We have added several of these excellent citations from a few more years back to our introduction and conclusion in order to provide a more robust framing of our analysis of competency-based training. 

Line 140: Why were the other articles rejected? Table 2 rightly includes some non-U.S. authors and context as global health is a global practice. What were the criteria for including these and perhaps rejecting others?

Response: Thank you a specific sentence on inclusion/exclusion has been added to our methods, “Articles were included if the presented discrete global health competencies or the implementation of a competency global health training program. Conversely articles were rejected if they did not contain data on the evaluation or implementation of competency-based curricula – i.e. they may have proposed a set of domains to cover, but did not have supporting information on the learner, curriculum delivery or assessment, etc.”

Line 147: How were these identified? is this a systematic methodology or eyeballing?

Response: Thank you for this helpful question. We have clarified the approach. 

Page 17 (no line numbers): The abstracted noted only 2 types of assessments.

Response: Thank you. We have revised to ensure alignment in the abstract. 

Page 17: Define "360 evaluation" for readers or provide reference.

Response: Thank you. We have added a more descriptive explanation and a reference. 

Page 19, second paragraph: This statement about public health is likely to be met with deep resistance. I do not think that most public health professionals will accept the view that public health thinking was build on a foundation of clinical settings.

Response: Thank you, and we have clarified our meaning in a revised opening to that paragraph.

---

## [Decision Letter · Decision Letter 1]

26 Aug 2020

PONE-D-19-35888R1

Implementation of Global Health Competencies:

A Scoping Review on Target Audiences, Levels, and Pedagogy and Assessment Strategies

PLOS ONE

Dear Dr. Schleiff,

Thank you for submitting your manuscript to PLOS ONE. After careful consideration, we feel that it has merit but does not fully meet PLOS ONE’s publication criteria as it currently stands. Therefore, we invite you to submit a revised version of the manuscript that addresses the points raised during the review process.

We look forward to receiving your revised manuscript.

Kind regards,

Elisa J. F. Houwink, MD, PhD

Academic Editor

PLOS ONE

Reviewers' comments:

Reviewer's Responses to Questions

**Comments to the Author**

1. If the authors have adequately addressed your comments raised in a previous round of review and you feel that this manuscript is now acceptable for publication, you may indicate that here to bypass the “Comments to the Author” section, enter your conflict of interest statement in the “Confidential to Editor” section, and submit your "Accept" recommendation.

Reviewer #3: All comments have been addressed

2. Is the manuscript technically sound, and do the data support the conclusions?

Reviewer #3: Yes

3. Has the statistical analysis been performed appropriately and rigorously? 

Reviewer #3: Yes

4. Have the authors made all data underlying the findings in their manuscript fully available?

Reviewer #3: Yes

5. Is the manuscript presented in an intelligible fashion and written in standard English?

Reviewer #3: Yes

6. Review Comments to the Author

Reviewer #3: (No Response)

7. PLOS authors have the option to publish the peer review history of their article (what does this mean?). If published, this will include your full peer review and any attached files.

Reviewer #3: **Yes: **Oladele A. Ogunseitan

---

## [Author Response · Author response to Decision Letter 1]

11 Sep 2020

Thank you for this opportunity to re-submit and move forward with the consideration of this article for publication. As discussed with the PLOS ONE team, there were no comments from the last round of review for us to address. If there is anything else we can do to help, please let us know. Very sincerely, Co-authors

---

## [Editor Report · Decision Letter 2]

16 Sep 2020

Implementation of Global Health Competencies:

A Scoping Review on Target Audiences, Levels, and Pedagogy and Assessment Strategies

PONE-D-19-35888R2

Dear Dr. Schleiff,

We’re pleased to inform you that your manuscript has been judged scientifically suitable for publication and will be formally accepted for publication once it meets all outstanding technical requirements.

Kind regards,

Elisa J. F. Houwink, MD, PhD

Academic Editor

PLOS ONE
---

## [Editor Report · Acceptance letter]

21 Sep 2020

PONE-D-19-35888R2 

Implementation of Global Health Competencies:
A Scoping Review on Target Audiences, Levels, and Pedagogy and Assessment Strategies 

Dear Dr. Schleiff:

I'm pleased to inform you that your manuscript has been deemed suitable for publication in PLOS ONE. Congratulations! Your manuscript is now with our production department. 

Kind regards, 

on behalf of

Dr. Elisa J. F. Houwink 

Academic Editor

PLOS ONE